# The Effectiveness and Safety of Stereotactic Body Radiation Therapy (SBRT) in the Treatment of Oligoprogressive Breast Cancer: A Systematic Review

**Bernie Yan** [1,†] , **Sherif Ramadan** [2,†] , **Katarzyna J. Jerzak** [3] , **Alexander V. Louie** [4] **and Elysia Donovan** [5,*]

1   Faculty of Health Sciences, McMaster University, Hamilton, ON L8S 4L8, Canada; yanb16@mcmaster.ca
2   Department of Radiation Oncology, London Health Sciences Centre, London, ON N6A 5W9, Canada; sherif.ramadan@lhsc.on.ca
3   Department of Medicine, Sunnybrook Odette Cancer Centre, Toronto, ON M4N 3M5, Canada; katarzyna.jerzak@sunnybrook.ca
4   Department of Radiation Oncology, Sunnybrook Health Sciences Centre, Toronto, ON M4N 3M5, Canada; alexander.louie@sunnybrook.ca
5   Department of Radiation Oncology, Juravinski Cancer Centre, Hamilton Health Sciences, Hamilton, ON L8V 5C2, Canada
*   Correspondence: elysia.donovan@gmail.com
†   These authors contributed equally to this work.

**Abstract:** Stereotactic body radiotherapy (SBRT) has emerged as a technique to treat oligoprogressive sites among patients with breast cancer who are otherwise doing well on systemic therapy. This study systematically reviewed the efficacy and safety of SBRT in the setting of oligoprogressive breast cancer. A literature search was conducted in the MEDLINE database. Studies regarding SBRT and oligoprogressive breast cancer were included. Key outcomes of interest were toxicity, local control, progression, and overall survival. From 863 references, five retrospective single-center cohort studies were identified. All studies included patients with both oligometastatic and oligoprogressive disease; 112 patients with oligoprogressive breast cancer were identified across these studies. Patient age ranged from 22 to 84, with a median of 55 years of age. Most patients had hormone-receptor-positive and HER2-negative disease. SBRT doses varied from 24 to 60 Gy in 1–10 fractions based on the location/size of the lesion. Forty toxicity events were reported, of which the majority (*n* = 25, 62.5%) were grade 1–2 events. Among 15 patients who received SBRT concurrently with a CDK4/6 inhibitor, 37.5% of patients experienced grade 3–5 toxicities. Progression-free and overall survival ranged from 17 to 57% and 62 to 91%, respectively. There are limited data on the role of SBRT in oligoprogressive breast cancer, and prospective evaluation of this strategy is awaited to inform its safety and efficacy.

**Keywords:** oligoprogression; SBRT; breast cancer

## 1. Introduction

Oligometastatic disease, defined as the presence of a limited number of metastatic lesions, is a growing area in the context of breast cancer [1]. Alongside the development of more effective and well-tolerated systemic treatments, the concept of a related condition called "oligoprogression" has also emerged [2]. Oligoprogression occurs when a limited number of metastatic lesions progress with otherwise controlled systemic disease while patients are on surveillance or receiving systemic therapy. [3] There remains no definitive consensus regarding the precise number of metastases that would distinguish oligoprogression from more extensive disease progression, although the most common definition involves three or fewer, with some studies including patients with up to five sites of disease progression [1,3]. From a therapeutic perspective, locally ablative therapy may provide an opportunity to regain control of metastatic disease in both the oligometastatic and oligoprogressive states [2]. For both scenarios, there has been a growing interest in the use

of focused radiation therapy at ablative dose levels, including stereotactic body radiation therapy (SBRT), as a form of metastasis-directed therapy (MDT) [4].

SBRT for oligoprogression has been evaluated in a limited number of settings. In prostate cancer, retrospective cohort studies have demonstrated that radiotherapy can prolong next-line systemic therapy-free survival (NEST-FS) and progression-free survival (PFS) among at least a subset of patients with oligoprogression [5]. A retrospective analysis similarly found that SBRT may have the potential to extend the duration of current systemic therapies among patients with renal cell carcinoma [6]. This could preserve subsequent treatments for later administration and, therefore, enable a longer total duration of treatment and possibly even a longer survival time [6]. However, even among disease sites that have been investigated, there remains a paucity of prospective data on the effectiveness of radiotherapy in treating oligoprogression [5,6]. Recently, a prospective phase II trial was carried out among patients with renal cell cancer receiving SBRT and tyrosine kinase inhibitor (TKI) therapy [7]. This study looked at thirty-seven oligoprogressive renal cell carcinoma patients and found that SBRT led to a 1-year local control of 93% and a median of 1-year delay in the need to switch systemic therapy treatment. Finally, the CURB study investigated SBRT among patients with oligoprogressive breast and lung cancer and was reported in abstract format at ASTRO 2021. This interim analysis demonstrated that SBRT led to improved progression-free survival in oligoprogressive lung cancer but did not show a similar result in breast cancer patients.

There remains a clear void in the scientific literature regarding the optimal method of treating patients with oligoprogression, whether stereotactic radiotherapy would provide clinical benefit, and in which patients the greatest benefits may be achieved. The aim of this study is to systematically review the safety and effectiveness of using SBRT to treat patients with oligoprogressive breast cancer.

## 2. Materials and Methods

A systematic review of the literature was performed according to the PRISMA systematic review guidelines (www.prisma-statement.org). The Ovid MEDLINE database was searched for relevant papers between 1946 and 2021 that met the study inclusion/exclusion criteria. To capture articles not indexed with MeSH, an additional search was performed on keywords occurring in the text of potential articles. These keywords included oligoprogression, oligometastasis, and oligorecurrent/residual disease and their synonyms. Additionally, the search included SBRT, stereotactic radiosurgery (SRS), stereotactic ablative radiotherapy (SABR), and synonyms of the techniques. A full list of the keywords can be seen in Figure 1. Searches were limited to publications in the English language.

### 2.1. Literature Search Strategy

Potential articles were identified using the National Library of Medicine's (NLM) medical subject headings (MeSH): "breast neoplasms AND radiosurgery".

### 2.2. Inclusion/Exclusion Criteria

We identified original reports including breast cancer patients with oligoprogressive disease (or "oligoprogression", defined as the progression of a limited number of metastatic sites with otherwise controlled disease on systemic therapy), who were treated using SBRT. Reports were excluded if they did not mention oligoprogressive and/or oligoprogression in their titles or abstracts, if they did not include patients with breast cancer, and if they did not evaluate SBRT as a treatment method. Articles exclusively including patients with oligometastases or intracranial oligoprogressive disease were excluded. Only articles containing original research data, including prospective and retrospective data, were included.

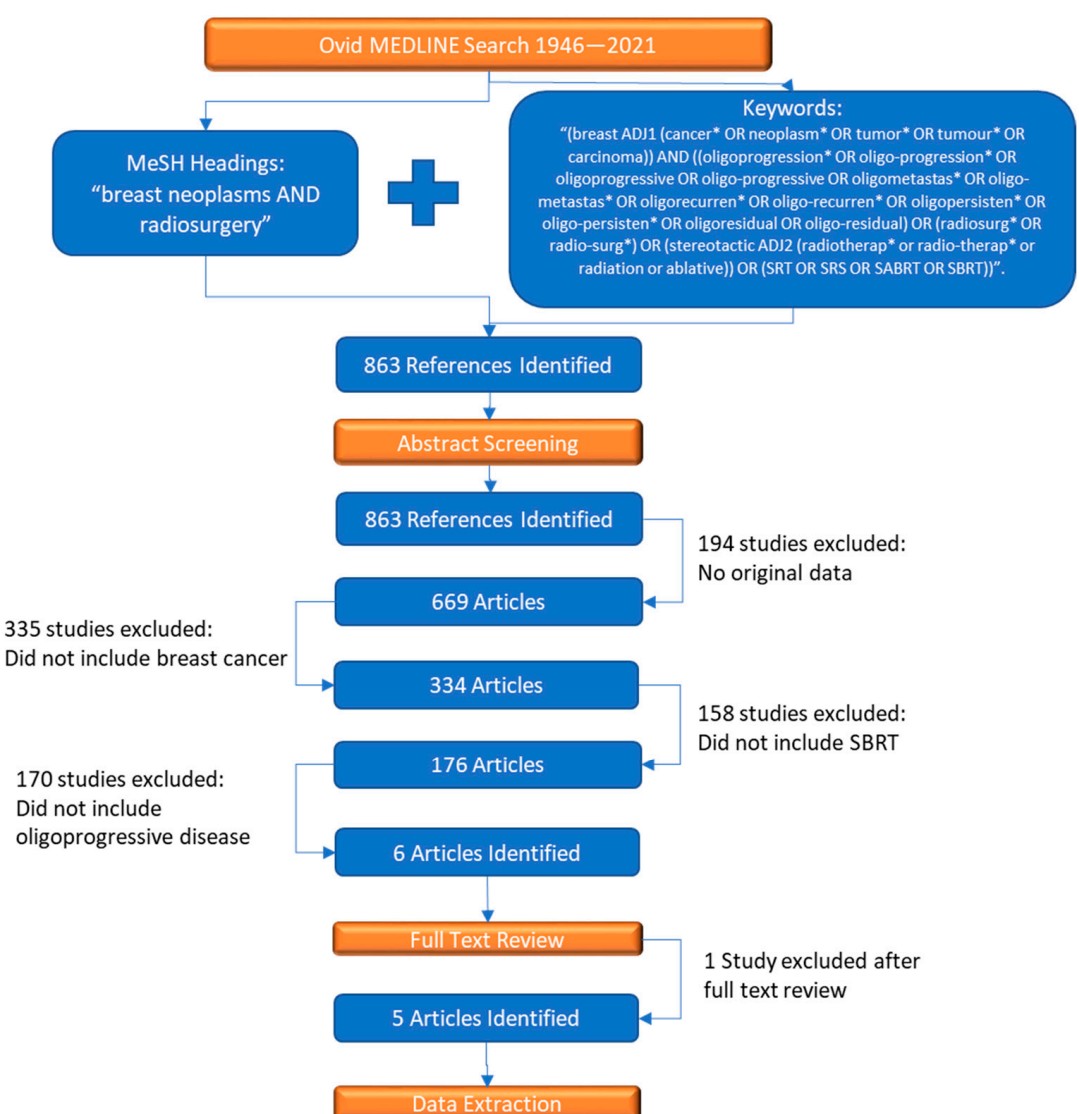

**Figure 1.** PRISMA flow diagram related to electronic search strategy. Five studies ultimately met criteria. Included studies were single-center retrospective cohort studies, with no prospective cohort studies nor randomized controlled trials identified. A total of 369 unique breast cancer patients were represented, and a total of 573 lesions were treated. There were 112 patients with oligoprogressive disease. Oligoprogressive disease was defined as less than or equal to five metastases.

### 2.3. Data Extraction

The following information was extracted from included reports: study design, study objectives, definition of oligometastases or oligoprogression used by each study, patient demographic information, disease characteristics, breast tumour subtype, time from diagnosis to SBRT, number of lines of systemic therapy, number of treated metastases per patient, location of metastatic sites, SBRT dose, and outcomes assessed. Reporting of oligometastatic and oligoprogressive data together versus separately in individual studies was also reviewed.

### 2.4. Outcomes of Interest

This study measured multiple oncologic outcomes associated with treatment using SBRT, including local control (LC), progression-free survival (PFS), and overall survival (OS). LC is defined as an absence of local recurrence at the treated site [8]. PFS is defined as the time from start of treatment until first evidence of tumour progression (either local or

distant) or until death from any cause, whichever event occurs first [9]. OS is defined as the time from start of treatment to death from any cause [9].

Toxicity was also assessed to determine if SBRT was tolerated among the study populations [10–12] following the Common Terminology Criteria for Adverse Events (CTCAE) on a scale of one to five. Descriptive analyses were performed to summarize study findings [13,14].

## 3. Results

*Literature Search Results*

The search identified 863 potentially eligible articles, and 6 studies met eligibility for analysis. A study by Shahi et al. was omitted after full text review [15]. A full PRISMA flow diagram can be seen in Figure 1.

All five studies contained a mixed patient population of both patients with oligo-progressive and oligometastatic disease. Across all studies, 34.9% of all patients had oligoprogression, with the range of patients with oligoprogression represented in each study varying from 12.5% to 67.3%. The full extracted set of data can be seen in Table 1.

Patient age ranged from 22 to 84 (median 55) years. Most patients with breast cancer had hormone receptor+/HER2-negative (HER2-) disease (mean 84%; range 71.8–100%). A study by Ippolito et al. exclusively included patients with HR+/HER2− disease. Patients received between one and four lines of systemic therapy prior to SBRT. Across the 573 metastatic sites, 56% were bone, 16% lung, 16% liver, and 12% represented another site.

SBRT doses varied from 24 to 60 Gy in 1–10 fractions based on the location and size of the lesions. $BED^{10}$ values varied from 60 to as high as 175. Three studies reported on toxicity using the Common Terminology Criteria for Adverse Events (CTCAE) reporting system. There were 40 toxicity events reported, with 62.5% ($n = 25$) grade 1–2 events and 37.5% ($n = 15$) grade 3–5 events. Toxicity outcomes are summarized in Table 2.

Three studies reported 2-year PFS rates, three reported 2-year OS rates, and two reported 2-year LC rates (Table 3). Two-year rates ranged from 87% to 89% LC, 17% to 57% PFS, and 62% to 91% OS, respectively (Table 3). The median PFS ranged from 11 to 33 months. These data include patients with both oligoprogressive and oligometastatic disease, as results were not reported separately among all studies.

An important note when discussing progression-free survival is the routine follow-up strategies utilized to detect progression of disease post-treatment. This is not explicitly stated in all of the five key studies identified in this systematic review. However, Weykamp et al. [11] noted that patients received CT for chest, abdomen, and pelvis every 3 months in follow-up post-SBRT treatment. This is similar to the strategy used by Tan et al. [14], where CT was used on a two- to four-month follow-up schedule, but they allowed for the addition of MRI if needed. Kelly et al. [13] also followed a two- to four-month follow-up schedule or triggered imaging on the presentation of patient symptomatology. Variability in follow-up imaging in this patient population also presents challenges when comparing the outcomes of SBRT to systemic therapy alone for oligoprogressive breast cancer.

Weykamp et al. performed a multivariate analysis and found that two-year PFS and OS rates were not significantly different between those with oligoprogression and those with oligometastases [11]. However, 2-year LC was noted to be 60% for oligoprogressive disease versus >95% for oligometastatic disease. Tan et al. reported separately on patients with oligoprogression versus oligometastases. They found two- year PFS rates of 52% and 8% and OS rates of 82.5% and 57.5% for patients with oligometastases and oligoprogression, respectively [14].

Quality of life (QOL) endpoints were not formally reported among the included studies.

**Table 1.** Summary of data abstracted from the included studies. NR: Not Reported, OP= Oligoprogressive, OM = Oligometastatic.

| Study | Study Design | Patient Characteristics | | | | Number of Patients | | | | Number of Patients | |
|---|---|---|---|---|---|---|---|---|---|---|---|
| | | Age | Tumour Subtype | Lines of Systemic Therapy | Metastatic Sites | OM | OP | Other | Total | Previous Chemotherapy | Previous Endocrine Therapy |
| Ippolito et al. [10] | Single-Centre Retrospective Cohort Study | Median 54 Range 30–80 | HR+/HER2−: 16 (100%) | NR | *n* = 24 lesions: Bone: 22 (91.6%) Other: 2 (8.4%) | 3 | 2 | 11 | 16 | 2 (12.4%) | 3 (18.7%) |
| Weykamp et al. [11] | Single-Centre Retrospective Cohort Study | Median 55 Range 27–82 | HR+/HER2−: 28 (71.8%) HER2+: 8 (20.5%) TNBC: 5 (6%) | NR | Bone: 19 (32.8%) Lung: 19 (32.8%) Liver: 19 (32.8%) Other: 1 (2.0%) | 32 | 14 | 0 | 46 | 33 (71.7%) | NR |
| Ari Wijetunga et al. [12] | Single-Centre Retrospective Cohort Study | Median 56 Range 30–83 | HR+/HER2−: 66 (84%) HER2+: 8 (10%) TNBC: 5 (6%) | 0: 16 (20%) ≥1: 63 (80%) | *n* = 103 lesions: Bone: 96 (93%) Lung: 2 (2%) Other: 5 (5%) | 42 | 37 | 0 | 79 | NR | NR |
| Tan et al. [14] | Single-Centre Retrospective Cohort Study | Median 54.8 Range 25–82 | HR+/HER2−: 99 (82.5%) HER2+: 21 (17.5%) | 0: 10 (5.2%) 1: 50 (25.9%) 2: 71 (36.8%) 3: 27 (13.9%) >4: 33 (17.2%) Unknown: 2 (1%) | *n* = 193 lesions: Bone: 113 (58.5%) Lung: 36 (18.7%) Liver: 39 (20.2%) Other: 5 (2.6%) | 66 | 36 | 18 | 120 | NR | NR |
| Kelly et al. [13] | Single-Centre Retrospective Cohort Study | Median 55 Range 22–84 | HR+/HER2−: 87 (81%) HER2+: 21 (19%) | 1: 57 (53%) 2: 18 (17%) 3: 29 (27%) 4: 4 (4%) | Bone: 69 (64%) Lung: 37 (34%) Liver: 34 (31%) Other: 55 (51%) | 11 | 23 | 74 | 108 | 45 (42.0%) | 44 (41.0%) |

**Table 2.** SBRT-associated toxicity among the six included studies.

| Study | Toxicity Levels Reported (Number of Patients) | |
|---|---|---|
| | Mild–Moderate (Grade 1–2) | Severe (Grade 3–5) |
| Ippolito et al. [10] | Haematological: 10 (62.5%) Non-Haematological: 5 (31.3%) | Haematological: 9 (56.3%) Non-Haematological: 1 (6.3%) |
| Weykamp et al. [11] | 10 (17.2%) | 0 (0%) |
| Tan et al. [14] | NR | 5 (4.2%) |
| Ari Wijetunga et al. [12] | NR | NR |
| Kelly et al. [13] | NR | NR |

NR: Not Reported.

**Table 3.** Two-year survival outcomes associated with treatment using SBRT [10–14].

| Study | Survival Outcomes | | | | |
|---|---|---|---|---|---|
| | 2-Year LC | 2-Year PFS | Median PFS | 2-Year OS | Median OS |
| Weykamp et al. [11] | 89% | 17% | NR | 62% | NR |
| Ari Wijetunga et al. [12] | NR | 57% | 33 Months | 91% | 86 Months |
| Tan et al. [14] | 87% | 32% | 11 Months | 70% | 53 Months |
| Ippolito et al. [10] | NR | NR | NR | NR | NR |
| Kelly et al. [13] | NR | NR | NR | NR | NR |

NR: Not Reported.

## 4. Discussion

This systematic review identified five retrospective studies investigating patients with both oligometastatic and oligoprogressive breast cancer. Overall, patients demonstrated promising LC, PFS, and OS after SBRT, suggesting that it may be an efficacious treatment option. SBRT was relatively well-tolerated among most patients. Weykamp et al. observed no grade 3 or higher toxicities at first follow-up [11]. However, in patients who received SBRT while on CDK4/6 inhibitors, a high incidence of grade 3, 4, and 5 haematological toxicities were seen. These toxicities are likely attributable to concurrent usage of CDK4/6 inhibitors, which are known to cause bone marrow suppression [10]. This highlights the importance of carefully considering and evaluating both the synergy and toxicity risk of SBRT with novel systemic therapies, which is an active area of investigation [16,17]. Additionally, Ari Wijetunga et al. observed that, 2–6 months after SBRT, 88% of oligometastatic and oligoprogressive breast cancer patients experienced improvements in their symptomatic metastases [12].

Patients with oligometastases displayed superior local control and PFS following SBRT versus those with oligoprogression in the two studies that analysed these disease states separately [11]. A review study by Saeed [18] summarized various retrospective and prospective studies on SBRT for oligometastatic breast cancer patients. They found that for patients with oligometastatic disease, 2-year LC varied from 66.1 to 100%, 2-year OS varied between 57 and 85%, and 2-year PFS from 8% to 53%. The authors suggested that patients with oligoprogressive disease may harbour more biologically aggressive and treatment-resistant tumours as compared to those with oligometastatic disease [11,14]. This is consistent with other reports in the literature, where oligometastatic breast cancer, especially confined to bone, may be characterized by an indolent disease course [18]. Weykamp et al. [11] noted that patients who had received chemotherapy prior to SBRT had worse LC outcomes. The authors again postulated that oligoprogressive disease biology may be more biologically aggressive and resistant. Patients with oligoprogressive breast cancer may, in turn, require higher doses of SBRT to achieve similar local control outcomes to patients with oligometastatic disease. The current studies utilize a wide variety of SBRT doses and subsequent BED$^{10}$ delivered to patients. This makes it difficult to analyse whether dose escalation would overcome the difference in disease biology and improve the LC and PFS for oligoprogression. A retrospective study by Nicosia et al. [19] was published

after this study's search strategy, and they looked at seventy-nine oligoprogressive breast cancer patients who had one hundred and fifty-three metastases treated. They found a median time to delay the next systemic therapy of eight months, and a two-year freedom from local progression of 86.7%. The time to polymetastatic conversion was 10 months, and the median OS was 72 months. They found that local control was related to disease progression and predictive of the likelihood to convert to polymetastatic disease. The potential for metastasis to spread and lead to polymetastatic states was noted to be a factor in favour of SBRT to delay disease progression. The median BED in this study was 78. If the BED was greater than seventy, the freedom from local progression was 90% versus 74.2% if the BED was <70. This is a lower threshold than a BED of 100 that was previously investigated and suggests that lower doses of radiation with reduced toxicity profiles could be investigated in the future. This study redemonstrates the importance of appropriate BED SBRT treatments in the oligoprogressive setting.

The efficacy of SBRT for oligoprogression may also be related to certain breast cancer disease factors. A majority of patients identified within this study had HR+/HER2-breast cancer. This cohort are more likely to present at advanced age, have early-stage disease, present with small tumours, and have a well/moderately differentiated histological grade, as compared to other breast cancer subtypes. All these features favour a more indolent disease course and potentially more favourable PFS and OS [18,20]. Patients with HR+/HER2-negative metastatic breast cancer are known to have survival times that exceed 5 years when receiving front-line endocrine therapy with a CDK4/6 inhibitor [20]; given that this oral endocrine therapy is generally well tolerated, the full potential of SBRT in this scenario may not be realized. However, oncologists should be aware that CDK4/6 inhibitors have the potential to increase the toxicity of SBRT. Preclinical studies have suggested that CDK4/6 inhibitors have a radio-sensitizing effect, with more than 10% of severe toxicities in previous clinical studies occurring in radiation fields. However, the majority of available research on the safe combined use of CDK4/6 inhibitors with radiotherapy has been conducted with limited retrospective data [21]. An expert taskforce led by Leblanc et al. looked at expert opinions and evidence regarding this topic. The report recommended the approach of pausing chemotherapy and CDK4/6 inhibitors during radiotherapy treatments [22]. As seen in the study by Ippolito et al. [10], there is a risk for significant toxicity with CDK4/6 inhibitors; however, it was felt that hormonal therapies and trastuzumab should be continued. A meta-analysis by Becherini et al. [23] investigated the safety profile of these CDK4/6 inhibitors when combined with radiation therapy. This meta-analysis looked at eleven retrospective studies, which included patients with early-stage and metastatic breast cancer who received concurrent radiotherapy and CDK4/6 inhibitors. Although there was heterogeneity in these studies, grade 3 or higher neutropenia was the most common side effect seen and was observed more commonly with palbociclib and ribociclib, as compared to abemaciclib. Other side effects included hematologic- or immune-mediated toxicity, including, enterocolitis, dermatitis, anaemia, thrombocytopenia, nausea, and vomiting. In particular, gastrointestinal toxicity was felt to be of significant concern with concurrent therapy. Several ongoing studies are investigating the use of concomitant CDK4/6 inhibitors and radiation with SBRT. Overall, the current level of evidence regarding the safety of concurrent radiation and CDK4/6 inhibition is low, with late toxicity, in particular, requiring further investigation. In an abundance of caution, general recommendations have been to hold these agents prior to the initiation of SBRT and for a short time after its completion [23].

The treatment of oligo-breast cancers remains a sparsely studied area, with evidence primarily extrapolated from other oligoprogressive disease sites. The BR002 study compared the efficacy of SBRT plus standard of care (SOC) versus SOC alone to treat patients with oligometastatic breast cancer whose disease was stable for less than or equal to 12 months on front-line systemic therapy [24]. Unfortunately, SBRT plus SOC was not found to provide significant therapeutic benefit for patients when compared to the SOC alone. However, this was a small trial of 129 patients, and almost 80% had HR+/HER2-

ve disease. Further, patients with oligoprogressive disease were not represented in this trial [20]. Additionally, the recent CURB study tested SBRT plus SOC therapies versus SOC alone among patients with oligoprogressive breast or lung cancers [25]. The primary endpoint was progression-free survival (PFS), and it was found at interim analysis that no statistically significant difference in median PFS was observed in the breast cancer cohort (18 weeks with SBRT + SOC compared to 17 weeks with SOC only) [25]. Conversely patients with lung cancer demonstrated significant benefit from SBRT (median of 44 weeks in the SBRT arm versus 9 weeks in the SOC arm, *p* = 0.004) [24]. This trial, however, included patients who had failed several lines of chemotherapy, and 32% (*n* = 15) of breast cancer patients had triple-negative disease, neither of which may be ideal candidates for SBRT [25]. The AVATAR trial recently opened to accrual in Australia, examining SBRT for patients with breast cancer, whose disease progressed on first-line CDK4/6 inhibitors. The study aims to enrol 32 patients with oligoprogressive HR+ and HER2– breast cancer. However, study findings will not be available for several years [4]. Finally, an expert committee opinion on the management of oligometastatic breast cancer [22] discussed potential approaches for these patients. It outlined that the choice of local therapy is dependent on patient age, level of comorbidity, and response to systemic therapy. It noted that SBRT can be efficacious for a variety of metastatic disease states. Further, the authors suggested there is no clear evidence to guide the optimal timing of radiotherapy with systemic treatments, and treatment decisions should be made on a case-by-case basis.

The limitations of this review include the small number of included studies and small number of patients with oligoprogression within these studies. Additionally, most studies did not report separate outcomes for patients with oligoprogressive or oligometastatic disease. No prospective trials were available, and, therefore, the quality of data is also limited to retrospective reviews. Finally, a range of breast cancer subtypes were included, with patients having received various lines and types of systemic therapy. This significantly impacts the ability to interpret results and determine which, if any, patients have the potential to benefit from SBRT for oligoprogressive breast cancer.

Further characterization of the optimal patient population and clinical scenario in which SBRT should be used is required. Potential populations of interest for further investigations may include patients with HER2+ or triple-negative disease. As systemic therapies for HER2+ breast cancer have been steadily improving, it is possible that the administration of SBRT may prolong time on any individual line of systemic therapy, with a potential to yield prolonged PFS and potentially even overall survival [26,27]. Those patients with triple-negative disease may also benefit from SBRT, despite their more aggressive disease biology due to a limited number of systemic therapy options. The potential need to hold systemic therapy to allow for SBRT needs to be considered, depending on the disease setting and type of systemic therapy being used. Finally, the impact of SBRT on QOL indicators and other patient-reported outcomes, when compared to that of the SOC, also requires further investigation. SBRT should be approached carefully in the treatment of oligoprogression, with consideration of the available systemic therapies. Further studies are required to determine which patient subgroups may benefit most from local ablation in this scenario.

## 5. Conclusions

There are currently insufficient data to conclude the efficacy of SBRT in managing oligoprogressive breast cancer. Based on the five included studies, SBRT provided good rates of PFS, OS, and LC and was well-tolerated among most patients, with related toxicities rarely resulting in long-term complications. The main toxicities that were noted occurred when SBRT was paired with CDK4/6 inhibitors. However, a significant limitation of the five identified studies is that each of the studies is a retrospective cohort study with relatively small sample sizes. Additionally, many of these studies did not separately analyse oligometastatic and oligoprogressive patients. Further prospective cohort studies and, ideally, randomized clinical trials should be conducted to assess the safety and efficacy

of SBRT to inform optimal clinical practice. Additional research is required before a definitive conclusion can be made on the use of SBRT in treating oligoprogressive breast cancer. These studies will hopefully address both the safety and efficacy of SBRT in this patient population.

**Author Contributions:** All authors contributed to this work. All authors have read and agreed to the published version of the manuscript.

**Funding:** This research received no external funding.

**Conflicts of Interest:** Alexander Louie discloses that he received speaker's fees and advisory board participation from AstraZeneca, unrelated to this project. Katarzyna Jerzak discloses the following potential COIs. She has consultant/advisory board member/honoraria for: Amgen, AstraZeneca, Apo Biologix, Eli Lilly, Esai, Genomic Health, Gilead Sciences, Knight Therapeutics, Merck, Myriad Genetics Inc., Pfizer, Roche, Seagen, Novartis, Viatris. She has received research grants from: AstraZeneca, Eli Lilly, Seage. Elysia Donovan: Sherif Ramadan, and Bernie Yan have no disclosures.

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
