# Peer review of "The Effectiveness and Safety of Stereotactic Body Radiation Therapy (SBRT) in the Treatment of Oligoprogressive Breast Cancer: A Systematic Review"

_curroncol, doi:10.3390/curroncol30070505_

Round 1

Reviewer 1 Report

The authors follow PRISMA methology for systematic review on topic of interest within the oligometastatic community. Manuscript is nicely written, summarizes pertinent data, and has relevant discussion of pertinent literature. Strengths and limitations are also appropriately laid out.

Author Response

Thank you for this comment.

Reviewer 2 Report

The manuscript is a systematic review about the effectiveness and safety of SBRT in the treatment of oligoprogressive breast cancer.

Evidences are particularly needed for this topic because no prospective data were available and the quality of retrospective studies is limited.

I have just few comments:

1.             Introduction

Page 2 lines 62/63: In the interim analysis of CURB study SBRT is associated with the improvement of PFS only in oligoprogressive lung cancer, as correctly reported in discussion. Please rephrase.

2.             Results

Page 5 lines 136: “Scheme 24. Gy in 1-10 fractions based on the location and size of the lesions”. This sentence is unclear. I suggest to use the phrase reported in the abstract: “SBRT doses varied from 24-60 Gy in 1-10 fractions based on the location and size of the lesions”

3.             Discussion

Page 6 lines 170: I suggest to add data from this recently published meta-analysis:

Safety profile of cyclin-dependent kinase (CDK) 4/6 inhibitors with concurrent radiation therapy: A systematic review and meta-analysis.

Becherini C, Visani L, Caini S, Bhattacharya IS, Kirby AM, Nader Marta G, Morgan G, Salvestrini V, Coles CE, Cortes J, Curigliano G, de Azambuja E, Harbeck N, Isacke CM, Kaidar-Person O, Marangoni E, Offersen B, Rugo HS, Morandi A, Lambertini M, Poortmans P, Livi L, Meattini I. Cancer Treat Rev. 2023 Jun 15;119:102586. doi: 10.1016/j.ctrv.2023.102586. Online ahead of print.

Author Response

The manuscript is a systematic review about the effectiveness and safety of SBRT in the treatment of oligoprogressive breast cancer. Evidences are particularly needed for this topic because no prospective data were available and the quality of retrospective studies is limited.

Reviewer Comment 1: Introduction : Page 2 lines 62/63: In the interim analysis of CURB study SBRT is associated with the improvement of PFS only in oligoprogressive lung cancer, as correctly reported in discussion. Please rephrase.

Author Response 1: Thank you for this comment. The statement on lines 62/63 has been rephrased to the following:

This interim analysis demonstrated that SBRT led to improved progression free survival in oligoprogressive lung cancer but did not show a similar result in breast cancer patients.

Reviewer Comment 2:   Results: Page 5 lines 136: “Scheme 24. Gy in 1-10 fractions based on the location and size of the lesions”. This sentence is unclear. I suggest to use the phrase reported in the abstract: “SBRT doses varied from 24-60 Gy in 1-10 fractions based on the location and size of the lesions”

Author Response 2: As recommended the statement has been changed to:

SBRT doses varied from 24-60 Gy in 1-10 fractions based on the location and size of the lesions.

Reviewer 3 Report

This well-written study “The effectiveness and safety of stereotactic body radiation therapy (SBRT) in the treatment of oligo progressive breast cancer: a systematic review” has a considerable clinical value.

However, I have following comments:

-          From the 5 studies presented in the manuscript, is it possible to determine how the presence of oligo-metastatic disease was detected and what imaging modality was used to assess progression? Please discus this issue briefly.

-          Some abbreviations are not defined in the text, e.g. QoL, SOC, etc. Please check this...

Author Response

This well-written study “The effectiveness and safety of stereotactic body radiation therapy (SBRT) in the treatment of oligo progressive breast cancer: a systematic review” has a considerable clinical value.

However, I have following comments:

Reviewer Comment 1:  From the 5 studies presented in the manuscript, is it possible to determine how the presence of oligo-metastatic disease was detected and what imaging modality was used to assess progression? Please discus this issue briefly.

Author Response 1: A brief discussion of this has been added to the Results section as follows:

An important note when discussing progression free survival is the routine follow-up strategies utilized to detect progression of disease post treatment. This is not explicitly stated in all of the five key studies identified in this systematic review. However, Weykamp et al [11] noted that patients had received CT chest, abdomen, and pelvis every 3 months in follow up post SBRT treatment. This is similar to the strategy used by Tan et al [14] where CT was used on a two to four month follow up schedule, but they allowed for the addition of MRI if needed. Kelly et al [13] also followed a two to four month follow-up schedule or triggered imaging on the presentation of patient symptomatology. Variability in follow up imaging in this patient population also presents challenges when comparing the outcomes of SBRT to systemic therapy alone for oligoprogressive breast cancer.

Reviewer Comment 2 Some abbreviations are not defined in the text, e.g. QoL, SOC, etc. Please check this...

Author Response 2: Thank you for this comment. These abbreviations have been corrected and clarified.